# Faculty Perspective Regarding Practical Experience of Adapted Physical Education for Undergraduate Students

**DOI:** 10.3390/ijerph20054282

**Published:** 2023-02-28

**Authors:** Seungyeon Park

**Affiliations:** Department of Health, Physical Education and Exercise Science, School of Education, Norfolk State University, Norfolk, VA 23504, USA; sypark@nsu.edu; Tel.: +1-757-823-8455

**Keywords:** adapted physical education, physical education teacher education, kinesiology, faculty perspective, higher education

## Abstract

Preparing future physical education professionals to teach adapted physical education (APE) is a responsibility of physical education teacher education (PETE) programs. Furthermore, there is limited literature on practicum and/or field experiences as a part of APE courses from the perspective of faculty members. The purpose of this qualitative study was to explore faculty views in relation to the practical experiences in undergraduate APE courses. Structured interviews were conducted with faculty members of higher education institutions in the U.S. There were five study participants in this study. Thematic analysis was employed for data analysis. The findings included three subthemes: (a) quality of quantity, (b) need for diversity in practical experiences, and (c) practical experience pertaining to APE courses. Practical experience in APE courses is an integral part of professional preparation for undergraduate students in kinesiology programs. There are no exact criteria for requirements across the states; however, students could maximize their learning by engaging in diverse APE practicum settings. The instructor of APE courses should provide clear guidelines and feedback for students. Instructors of APE courses must also consider the institutional and environmental context prior to planning and implementing practical experiences to provide successful learning experiences for students.

## 1. Introduction

In PETE and kinesiology programs, APE courses often provide field experiences as an integral part to promoting student learning. Those practical experiences will strengthen the knowledge of students by allowing them to learn how to work with individuals with disabilities in diverse physical education and/or activity settings [1,2]. The findings of the literature indicate that practical experiences of APE courses have a positive influence on undergraduate students’ attitude and perspective related to APE. Particularly, practical experiences in APE courses allow students to develop favorable attitudes toward populations with disabilities [3,4,5,6]. Practical experience of an APE course is an integral part to enhance the pedagogical knowledge of undergraduate students. It is important to provide numerous quality practical experiences for undergraduate students so that they can learn how to plan and implement APE when they become in-service teachers [6,7]. To improve the effectiveness of student learning in APE courses, students should be reflective and think critically while learning how to organize and modify physical education for populations with special needs through practical experiences [8,9]. There are often limited course offerings in higher education that provide content regarding populations with disabilities in physical education or physical activity settings.

In general, one APE course is provided for undergraduate kinesiology and PETE programs at universities [10,11]. Approximately 84% of APE courses require practical experience as a part of course completion [11]. On average, practical experiences in APE courses require one and half to two hours of engagement per week for undergraduate students at universities [12,13]. To become a certified adapted physical educator (CAPE), additional practical experiences and courses are required. CAPE teacher candidates are required to complete a minimum of 200 h in physical education or activity with individuals with disabilities [14]. However, in many areas, teachers do not have to be a CAPE to teach an APE class.

Practical Experiences and APE. There are several options for practical experiences in an APE course. Practical experience can take place in a gym or other physical activity facilities at a university, in public schools, or in community settings. Practical experiences in public schools allow undergraduate students to observe and assist physical education teachers in APE classes or special education teachers. Undergraduate students can learn the basic components of APE in public school settings in other practical ways. For instance, undergraduate students could enhance their understanding regarding the components of APE within an Individualized Education Plan (IEP) by closely interacting with in-service teachers. Service learning in community settings such as special Olympics and disability sporting events enables undergraduate students to broaden their perspective regarding physical activity and disability beyond APE in the school context. For this reason, to strengthen the learning of undergraduate students, practical experiences in APE courses need to be conducted in diverse environments including both inclusive settings and self-contained contexts [1].

Teacher education programs often prepare their curriculum by aligning with national and state standards. PETE programs organize their own curriculums by reflecting national- and state-level professional standards such as Adapted Physical Education National Standards (APENS) and the National Association of Sport and Physical Education (NASPE) [15,16]. There are Adapted Physical Education National Standards (APENS) that provide directions for PETE programs with respect to knowledge for working with students with special needs. The National Council for Accreditation of Teacher Education (NCATE) standards are also used to guide PETE programs [17]. Under the NCATE, the NASPE provides the specific parameters for PETE program development, such as the content areas pre-service teachers are required to gain knowledge in. For instance, there is a specific component that addresses the role of pre-service teachers and students with disability (i.e., Standard 3. Planning and Implementation—“effective resources, accommodations and/or modifications, technology and metacognitive strategies to address the diverse needs of all students”) [18]. Meeting the “diverse needs of all students” is achieved by enhancing the knowledge and practical experiences of pre-service teachers so that they can be equipped with the skills to accommodate students with unique needs and provide inclusive environments. Regretfully, many undergraduate students in PETE programs expressed that they had hesitation or felt unprepared when they met students with special needs [19,20]. This could be attributed to the limited number of required courses in PETE programs that allow students to work with populations with special needs. Thus, many in-service teachers lack professional training for working with populations with disabilities.

There is a line of research that addresses the benefits of practical experiences in APE. Students who engaged in practical experiences were more likely to have favorable attitudes toward populations with special needs [20,21,22,23,24]. Students in APE courses enhanced their understanding about populations with disabilities and pedagogical knowledge through practical experiences [21]. In general, there was no significant difference in impact based on the type of APE practicum setting (e.g., campus-based and community-based practical experience). This suggests that both types of experiences improved undergraduate students’ confidence for working with populations with disabilities [1,12]. The literature indicates the importance of the quality of practical experiences in APE courses. There is also variability between faculty members and how they plan and implement practical experiences in their APE course. In a recent meta-analysis review, the authors described those practical experiences in the community setting, with numerous self-choice options being most beneficial for changing attitudes of undergraduate students toward populations with disabilities [25]. In addition, depending on the specific major (e.g., PETE, Kinesiology, Exercise Science) within the program, there were different impacts on the attitude of students toward individuals with disabilities. Although there is a depth of research pertaining to practical experiences in APE courses, much of the research is from the student’s view. There is a lack of attention paid to the perspective of university faculty, which is critical considering the faculty design and implement the practical experiences. Therefore, the current research explored perspectives of faculty who teach APE courses for various institutions in the field of kinesiology and PETE programs.

## 2. Materials and Methods

Given the limited research, the purpose of the current study was to explore the perspectives of faculty in relation to the practical experiences in APE courses for undergraduate students in PETE and kinesiology programs. Specifically, there were two research questions: How do various factors influence the planning and implementing of practical experiences in APE courses for undergraduate students? How do faculty effectively create practical experiences in an APE course to enhance undergraduate students’ knowledge and understanding of working with populations with disabilities?

### 2.1. Recruitment

This study used a qualitative design to explore the perspectives of faculty regarding practical experiences in APE courses for undergraduate students. Using a purposive sampling method, study participants were recruited from higher education institutions in an eastern area of the United States [26]. All the participants have working experience at diverse institutions. Table 1 summarizes the demographic information of study participants. When the interviews were conducted, all the participants were mainly working for Historically Black Colleges and Universities (HBCUs) in the field of kinesiology. An APE course is required for students majoring in physical education and is available as an elective course for special education majors at the institutions where participants worked. Generally, students register for APE courses during their junior and senior year after completing freshmen and sophomore academic year prerequisites. With an approval from the Institutional Review Board, consent forms were obtained from each study participant prior to data collection. Pseudonyms were assigned to each participant to protect their identity. Due to the COVID-19 pandemic, both online video meetings and in-person meetings were conducted. Online video meetings were preferred for two participants (Participants 2 and 3). Other participants had in-person interviews by maintaining social distancing and complying with COVID-19 health policies of their respective institutions (Participants 1, 4, and 5).

### 2.2. Participants

The aim of the current study was to explore perspectives of faculty regarding practical experiences of APE courses for undergraduate students in the department of kinesiology. The ultimate goal of the study was to explore the ways in which faculty provide undergraduate students with quality practical experiences in APE. There are common features among study participants. All participants completed their doctoral degree at one of the leading doctoral degree universities in the U.S. They had graduate assistant positions when they were registered in their doctoral programs. Currently, they are all tenured associates or full professors. One participant (3) is currently working as an adjunct professor since she retired. Each participant had different work experiences. Three faculty members (3, 4, and 5) had full-time faculty working experience at both PWI and HBCUs, and two participants have worked in HBCUs (1 and 2). Four participants of this study have been taught APE courses. More specifically, participant 2 is currently an associate professor mostly teaching APE and PETE coursework in a kinesiology program. She also serves in an administrative position and is on a university curriculum committee. She has faculty experience at two HBCUs after she received a doctorate in the field of kinesiology majoring in APE. Participant 1 is currently an associate professor in a kinesiology department and has previously served as department chair. He has instructed numerous subjects including APE. Participant 4 is currently an associate professor. She has also previously served as the department chair and is currently teaching APE and PETE coursework. She is involved in the curriculum development committee and has worked in professional organizations for APE and is highly involved in APE research. It was expected that their faculty and professional expertise would contribute knowledge for APE courses and practical experiences. Participant 3, now retired, had a specialization in motor skill development in the field of kinesiology. She combined theory and practice in her motor development courses and covered different age groups and populations with and without special needs. Participant 5 has expertise in the field of exercise science. She includes hands-on experiences for her classes focusing on anatomy and physiological courses and covers diverse learners with and without disabilities. She also has working experience at a rehabilitation and medical center.

### 2.3. Data Collection

Prior to participating in focused interviews, a demographic questionnaire was completed. The questionnaire collected information such as faculty experience, working years, type of institutions, and concentrated areas. The researcher of this study conducted individual, focused interviews with each participant. Interview questions were related to three guiding topics: (a) contextual factors that influence APE course delivery, (b) unique challenges of APE, and (c) teaching practices and instruction for APE for undergraduate students. One week ahead of the interview, questions were sent to each participant by e-mail. It was expected that interviewees could conceptualize their perspective and then reflect on the interview questions. The study was conducted after an approval from the Internal Review Board. Because of the COVID-19 pandemic, interviews were conducted either via a virtual meeting method (e.g., zoom) or in-person individual interview maintaining social distancing and complying with campus health protocols. The interviews took approximately 60 to 120 min (average 1.5 h). There were follow-up e-mail communications or direct meetings for several participants to clarify answers on interview questions or demographic information.

### 2.4. Data Analysis and Trustworthiness

There were several steps for data analysis. First, all the interviews were recorded (i.e., voice recorder, and online interviews based on computer). Next, the interviews were all transcribed. The researcher listened to the recorded files to transcribe the interviews verbatim and to ensure accuracy. A thematic approach was used to analyze the data. Common themes and subthemes emerged from the data. The common themes developed based on similar responses from study participants [27,28]. To identify the common themes, transcribed responses were divided into subthemes that represented the similar views from participants. All the subcategories were reviewed across study participants. Lastly, the researcher of this study checked for similarities and differences across the data allowing for the primary themes and subthemes to become apparent.

To ensure trustworthiness of the findings, the current study established the following four criteria: credibility, transferability, dependability, and confirmability—which are basic components of qualitative research [29]. To establish credibility, there were several strategies, including triangulation, member checking, and peer-debriefing in the current study. For triangulation, the recorded interviews for each participant and written transcripts were used to ensure the accuracy of the data. Member checking was completed to strengthen the credibility of the study’s findings to confirm the researcher’s interpretations. Further, the researcher reviewed the data to check for congruency during and after analysis. The researcher sent e-mails to study participants to confirm the accuracy of the interviewed data and interpretation. Peer-debriefing allowed for obtaining an external check regarding data interpretation, which increases credibility in the current study. For transferability, in the current study, the researcher provided an extensive description of the background and environment of the current study. By clearly describing research procedures and the methodology, this study demonstrates dependability.

## 3. Results

There were three themes that emerged from this study. The first theme, Quality of Quantity, is related to the quantity and quality of practical experiences needed for undergraduate students in APE courses. The second theme, Need for Diversity in Practical Experiences, reveals limitations and strategies to provide meaningful practical experiences in APE courses. The third theme, Practical Experience pertaining to APE course, describes how the APE course itself could strengthen practical learning for students. However, subthemes are interrelated to one another.

### 3.1. Quality of Quantity

Faculty members believed that practical experience in APE courses is critical for student learning. There were several common notions—“it is more about how than what”. Participant 3 expressed the key point of practical experience for APE courses. Fulfilling practicum hours as a part of course requirements is important; however, the key for student learning would be what they experience through those practicum hours.


*I don’t think more is better necessarily. Less hours with more supervision will be better for quality of hands-on experience. If you can go with your class, one time and see maybe two hours. Have them to talk about what they did. One time go and one time reflection time. How many different things need to talk about. Quality is important. I think supervising is important just rather than somewhere look around text using phone for instance.*


With required hours for practical experience in APE courses, she further pointed out the importance of the faculty role to support students learning in their practicum environment. Likewise, other study participants expressed similar viewpoints in that they highly value the importance of the faculty role for students’ practical experience. For instance, interaction with students was important for participant 1, rather than putting the students into a practicum. This process was critical for participant 1 to better understand the students’ perspective and provide corrective feedback for students to enhance their learning. For quality of practical experience in APE, participant 3 suggested that there are basic components that instructors need to address before they start any field experiences. She emphasized going back to the basics.


*Well, one, before students go into the field, they need to be aware of how they should present themselves. Know the expectation, how to dress, how to speak, how to know who their contact is.*


She emphasized the importance of developing a professionalism work ethic, showing respect to others, and establishing authority through their appearance and communication. Another participant, participant 4, shared her opinion regarding the criteria of practical experiences required in APE courses. Rather than merely explaining the criteria for the practical experience, she also highlights the importance of practical experiences in diverse settings in sequential ways.


*It doesn’t matter how many hours, there is no requirement there. The only hours required in PE is the state requirement for student teaching. There is no requirement before those hours. SHAPE wants, when they actually did recognition, they want you to have an experience at the middle school, or high school, and one in elementary. So, you needed to have at least two. I think to really get the students ready for internship they need four or five experiences. They do not all have to be long, but they do have to be different and sequential.*


As described above, she explained what a desirable direction for students in terms of their practicum and field experience would be, rather than merely connecting to a time-related factor for practicums. Participant 4 explained how practicum hours connected to the national standards and how she made changes by considering the institutional context. It was integral to include practicum experiences for undergraduate students aligning with the national standards. Concurrently, she explained about institutional context when planning to require practicum experience in a course. Institutional contexts were related to the location and the characteristics of the PETE program at the institution.


*I use the SHAPE America standards because our students with one course could never meet all those APENS (Adapted Physical Education National Standards). Now those APENS standards are built on the old NASPE (The National Association of Sport and Physical Education) standards which were turned into the Shape America standards. So, to me, yeah, it is applying those things they are supposed to be learning. Any interaction they can have, all of our students are planning to interact with others as their profession and they need to have more experience with that. I also think that can help as they get into their internship as they put interviews, adding to their resume, all of that. It is important. I will be honest with you, and most other institutions require twenty hours.*


She specifically described examples of how institutional context could have an impact on practical experience for students.


*Here this is probably one modification I have made I have done ten because transportation is a problem. And where our school is in a little bit of a desert. I mean there is houses here, and there are residential things here but there is no stores here, no recreation center. They could take the light rail… Students could take the light rail and go downtown but we are not even that close to schools other than the one high school up here where we often do the Special Olympics. Other than that, they can’t even walk to something there is no public transportation. So, I went down to ten hours, because even doing that ten hour hands-on that is probably going to be another 5 h just in transportation. And maybe five hours planning for the transportation by finding someone else who is going and all of that. If we had transportation, or close that they could walk to, then I would probably up it to 15 or twenty hours. Because ten hours is very minimal.*


Participant 4′s belief was that PETE students need to be exposed to more diverse practicum settings, while institutional context could be an influential factor. Study participants expressed that there is no right answer for the criteria of practicum requirements. Rather than emphasizing the requirements of practicums, they indicated that the quality of practical experiences matters especially when considering environmental characteristics and institutional context.

### 3.2. Need for Diversity in Practical Experiences

There are many factors that could impact the practical experiences of students in APE. For instance, participant 5 explained complicated situations that students could face when they have practical experiences in public schools. She pointed out the importance of networks to find the best options for students and their practical experience cited within given limited options for practical experiences for APE.


*So, when you are observing you are still limited because that class is blended with, it may be a regular Physical Education teacher that makes modification for one or two students within a larger class. But it really depends on the school district. So, you are at the mercy of, or the placement of that student what school district and how the school district is set up as far as if they even have Adapted Physical Educator and those Adapted Physical Education classes to really learn at that moment where they are involved in the course, or where they are involved in their student teaching experience.*


She implied that it will be important to carefully choose APE practicum places in k-12 in advance so that practicum experience of PETE students could vary and be different depending on school districts. Participants described diverse factors that could impact options for practical experiences for students. These factors were mostly related to the physical environment. They could learn how to use assessment tools, and how to modify and accommodate activities for their paired kids as a weekly assignment for one semester of an academic year that could enhance their practical learning in terms of pre-service teachers. The faculty view in this study was similar to this suggestion. For instance, participant 3 explained that practical experiences could be conducted in many different places as possible.


*Large schools, small schools, and diverse school settings in terms of race and ethnicity. All the different experiences. Students broaden their perspective more and more.*


In addition to the importance of experiencing as diverse settings as possible, participant 4 explained that students in APE courses may have different programs within kinesiology programs; thus, diverse practical experiences matching their specific program will be more appropriate for their career. She shared one approach to make students more active and accountable regarding their career development in APE field experience to match their areas.


*I give them a choice and they do their ten hours, but I also make them interview. So, when I used to do 20 h I required 10 h in the school for PE people and I required 10 h in a disability sport or recreation. That wouldn’t work for all our students. So, because they are not all going to be teachers. And the schools are not going or let them in even before the pandemic. So even those 10 h, before they would go out, I would require them to interview at least one person. So, if they were doing like, big feet meat (local Special Olympic) it would probably be Harris (anonymous name, APE K-12 teacher) or one of the other teachers or recreation people or parents who are there. And it is a very relaxed paper because it has more about their experiences and their thoughts how it is going to impact them.*


She described the importance of matching the major students into APE practicum settings that are more reasonable and allowed the students to be more proactive.

### 3.3. Practical Experience Pertaining to APE Course

All the interview participants agreed on the importance of practical experiences in an APE course for developing the knowledge of students and broadening their perspective toward populations with disabilities. The faculty in this study also emphasized the importance of the APE course itself as critical for student learning. Practical experiences are included as a supplemental part of APE courses to strengthen the knowledge of students. The APE course is only one course to learn about PE and Sports for populations with disabilities for many undergraduate kinesiology and PETE curriculums; therefore, the APE course can be new and not familiar to students. On a weekly basis during the semester, the APE course needs to deliver basic content knowledge for student learning. Based on the content knowledge, practical experiences during the class should enhance the practical learning of students. Participant 2 perceived the importance of practical experience embedded in APE courses.


*And then just really filling that course with a lot of examples, discussions and hands-on experiences about experience where you are being prepared in your program to help them really see how this is really just very different from regular physical education or regular course that they may teach because there is so many different factors that have to be considered with Adapted Physical Education that’s new.*


She explained that students should enhance their understanding of APE by learning different methods for APE instruction and management, and rules and regulations about APE because they are different from their regular physical education viewpoint. The effectiveness of APE courses was further explained by participant 2 who tried to encompass practical experiences on a weekly basis in her course.


*Hands-on experiences are needed, sometimes it is really hard though to find places where they can observe or work or volunteer and be a part of the school or program where they actually get hands-on experiences. But that’s why we have all those projects in our classes and when we do have a relationship with the school where they can get that we really hold onto it because it is really hard, it is becoming more and more harder to have great experiences where you are talking a student, a college student working and a Pre-K-12 setting where they actually get some hands on experience before they graduate with someone with a disability.*


Her perspective was that APE and field experiences could go hand in hand, when possible, but sometimes it is challenging. She describes that utilizing projects in the APE course could strengthen pedagogical knowledge of students, in addition to, or in some cases, in place of, practicum experiences. Similarly, participant 4 perceived that APE courses can include a certain range of practical experience for students. Specifically, exposure to diverse settings from an early phase at the university would be more helpful for students to build their career preparation.


*Some of the books still say NASPE. I wish there was still a NASPE. The practicum, I think it needs to be before the practicum. I think there needs to be a practicum in the class. But I think in most of our classes, there needs to be a practicum or a hands-on or an experience. It doesn’t always have to be 10 or 20 h (for practicum requirement). It can be smaller than that, but they have to do something. I think it is more meaningful to students as well. And also, if we want them to be leaders in their field, we need to be helping them build their resume while they are here and not wait until their internship.*


She explained that more diverse practical experience can be advantageous in terms practical learning and career development of students. APE courses are introductory and a new process for many students. On a weekly basis, instructors of APE courses can effectively provide foundational information about the specific type of disabilities students may encounter in PE or sport settings. Participant 4 explained:


*It is just so much to teach within that one course which is really limiting because there is not going to be another course that really goes into the specific disabilities… when it comes to Adapted Physical Education maybe they haven’t really considered the details that go into modifying a class for people with disabilities so this can be a completely new area to the students and so we have to not make any assumptions.*


Not only are APE courses new to students, but she also further described that even the prerequisite classes are often not related to APE.


*But the pre-requisites that prepare students for this class is not actually related to modification as far as disabilities in most of them. So, this may be the first time where they know that they body changes but of course there are so many disabilities… You may modify instructions for various skill levels in your course but then as far as disabilities that is a whole other element. So that’s one of the ways in which it differs as a course because you really have to cover a lot to give them that foundation of knowledge about laws, about the various disabilities, firs the ones that are prevalent in school systems and then just the IEPs, parent involvement, even like the places where instruction start at three years old instead of K-12.*


She felt it important to deliver the most important features of foundational knowledge regarding APE and only having one course makes it challenging to go in depth about a specific topic. The participants recognized that the APE courses at each institution can vary depending on the characteristics of the specific program and institution. Participant 1 explained the connection between institutional context and APE courses because the curriculum of each program will have a different emphasis such as therapeutic recreation, education, or rehabilitation.


*Yeah, and Wellington university (pseudonym) I think is more of a therapeutic rec. It serves that population too. So, I think it just depends and to me that’s what I see as more of a challenge than the students. And the difference with the students is I think our modalities of teaching need to be different. I’m adding more videos, adding shorter amounts of contact, uh, content. I’m doing more checks. (Like in the APE course I now have a quiz for every single chapter. But my quiz might only be 3–5 multiple choice questions. It’s very short.*


Additionally, interview participants shared that feedback on practical experiences would be beneficial. Students can be encouraged by instructors to be more accountable and active and to have more meaningful and effective practical experiences. To strengthen practical learning and critical thinking of students in APE courses, instructors can include requirements such as reflection of practical experience, discussions, and case study examples. Based on this, participant 1 explained that one of the roles of instructors is to provide feedback and have communication with students.


*I have conversations with them all the way through. I give a lot of feedback and frequent feedback. I also give a lot of prompts, but I have always done that. I try to have them be successful and give lots of opportunities for that, but I have always done that. I do a lot of overlap of content So if you missed it the first time, we’ll get it here… And engage them in learning. You can’t force them to learn, you have to engage them to learn. So, I try to do that.*


Another participant, participant 2, explained hands-on experience in the course and the role of the instructor.


*Hands on based learning, I taught back again in turn. They could not do it. Concept came through step by step. The same thing and same direction with hands on kind of concepts. When they go there, they see stuffs, you as the instructor or TA need to be there for sure as the instructor. For instance, I went there Tuesday with students, and back on Tuesday and have conversations with students. And tried to check each detail as possible.*


Across the interview, she repeatedly expressed that student learning can be enhanced with repetitive mastery and continuous check-ins. Beyond having practical experiences in APE courses, there would be more than that. One of the strategies could include a certain type of reflection so that students could be reminded of and relive their experiences. Participant 4 noted:


*But I think they also need to have time to reflect, and to learn how to reflect. So, I do a lot of that on discussion boards where they need to interact with each other. And it is a very relaxed paper because it has more about their experiences and their thoughts how it is going to impact them. I think I have 5 or 6 questions that are kind of open ended to lead them in that direction and then they have to finish their hours. They have a log of their hours. And they do the interview and then just kind of talk about all of it. And I have to say it is kind of an end of semester assignment even though they are doing it all during the semester the paper is toward the end and I have to say I have walked away with that paper going wow they did learn to think outside the box.*


To strengthen student knowledge and make them more critical learners, she believed reflective journaling, discussions, and an end-of-semester essay could be implemented during the semester. Table 2 includes the results of this study to represent the points to be considered when planning and implementing practical experiences of APE. Each subtheme with supporting comments and/or the points is described.

## 4. Discussion

The current study explored the perspective of faculty with relation to practical experiences in an APE course for undergraduate students in kinesiology and PETE programs. There were two research questions guiding this research: How do various factors influence the planning and implementing of practical experiences in APE courses for undergraduate students? How do faculty effectively create practical experiences in an APE course to enhance undergraduate students’ knowledge and understanding of working with populations with disabilities? There were themes that emerged from the participant interviews: (a) Quality of Quantity, (b) Need for Diversity in Practical Experiences, and (c) Practical Experience pertaining to APE course. The findings align with the previous literature that states that practical experiences are an integral part for student learning in terms of their pedagogical practice and professional preparation [20,21,22,23,24]. Particularly, based on the national standards (e.g., SHAPE America), it is recommended that undergraduate students should be exposed to diverse settings in terms or prek-12 and diverse leaners before student teaching because it will lead students to be more competitive and prepared [18]. The faculty emphasis on the need for diverse practical experiences in this study is consistent with previous literature on practical experiences in APE courses [23,30].

Furthermore, for practical experiences to be effective and of high quality, faculty have an important role in improving students’ learning during the process of practical experiences. From the very beginning, instructors must provide clear expectations for students about what is expected in terms of their professional behavior and engagement in practical settings [30]. These findings are consistent with previous studies that shed light on the role of active interaction with students during their practicum and course completion at the college level [31,32].

Contextual factors are another important variable that impact practical experiences. There are variables that could impact the options for practical experiences in the APE course. For instance, the structure and physical environment could be a decisive factor.

In addition, more diverse practical experiences would be beneficial for student learning. The options available to students for practical experiences will vary depending on location, physical environment, and other contextual factors of the institution. For instance, variables such as the lack of transportation and limited adapted physical sports and activities available in the community would negatively impact the overall practical experience.

With relation to the second question as to how faculty effectively create practical experiences in an APE course to enhance undergraduate students’ knowledge and understanding of working with populations with disabilities, there were several findings. Depending on the ways in which practical experiences were organized as a part of the APE course, the faculty in this study believed their students would experience professional growth. As discussed in the above, the faculty had a crucial role for practical experiences. It could be beneficial for students if they have on-campus practical experiences for APE courses and if students in the APE course have barriers to attending practical experiences, such as transportation issues and a lack of options in the community settings. There would be benefits because students in APE courses could learn each topic and then connect it to the weekly practicum in sequential ways. Students in the APE practicum can be paired with individuals with a disability. For instance, they could learn how to plan and implement an activity or lesson based on the specific type of disability. However, diverse options for practical experiences both on- and off-campus would still be beneficial for student learning [1,4]. A campus-based APE practicum is a self-contained environment. However, undergraduate students also need to experience inclusive settings because they can strengthen their knowledge pertaining to working with individuals with disabilities and become aware of real-word settings by going to community-based schools and facilities [1,12].

Generally, undergraduate students only take one APE course to learn how to work with populations with special needs in PE and physical activity settings. Students are likely to lack an understanding and be hesitated to work with populations with disabilities [30,33,34]. Therefore, APE courses should include a depth of information regarding the disabilities students may encounter. Once a foundation of knowledge is provided at an introductory level, students can be gradually introduced to more intermediate levels of learning through diverse hands-on experiences. This approach will allow students to have repeated mastery as well as enable instructors to observe the progress of students.

### Limitations and Future Directions

This study has several limitations. There should be caution to interpret the results of this study, because of the small size of the sample. However, the focus of this qualitative research was explored to gain an in-depth understanding of practical experience of APE courses from the view of faculty in the context of higher education institutions in the U.S. Another limitation is that two study participants had their expertise in the areas of PETE (motor development) and exercise science (athletic training). The researcher included these two participants by considering their diverse faculty experience. For instance, these two participants included several topics about individuals with disabilities in their curriculums. Furthermore, they provided practical experiences and/or hands-on experience for undergraduate students toward working with individuals with disabilities. Lastly, although the focus of this study was based on the view of faculty, investigating the perspective of diverse stakeholders of APE (e.g., parents or guardians of individuals with disabilities, and in-service and/or pre-service teachers) will be needed in that there are a limited number of these lines of inquiries. In addition, a future study should consider more contextual factors such as the types of disabilities, and diverse age groups connecting to APE practicum settings.

## 5. Conclusions

The current study explored the ways in which practical experiences in APE courses could be effectively conducted for practical learning of undergraduate students in the areas of kinesiology and PETE programs. There is no doubt that faculty believed pre-service teachers and practitioners can enhance their learning through diverse experiences. The findings of this study further explain how undergraduate students in APE courses could maximize their learning through practical experiences of APE courses. It is not enough to merely fulfil practicum hours as a requirement. The quality of practical experiences may be more important than the overall number of hours spent in practical experiences. There are very little criteria for student practicum hours or details specifically for APE courses; however, the faculty indicated that more diverse settings would certainly be beneficial. The findings explained the role of instructors in more specific ways. It is necessary to have active communication with students so that faculty can provide corrective feedback and then check their student’s learning, which could result in more successful and meaningful practical experiences. The faculty also indicated the importance of networking in the community to find the best options for practical experiences.

Contextual factors such as physical environment could be a decisive factor that could impact the options for practical experiences in the APE course. It could be appropriate to give a choice to students for practical experiences in the context of their specialization and program because there can be several different areas of focus in programs within the area of kinesiology. The faculty indicated that making a bridge for practical experience through APE lectures will be one effective way.

Lastly, there can be several different foci within kinesiology and PETE programs. Therefore, instructors of APE courses should consider the specific program focus to determine which experiences will be most meaningful and related to their specific major. By doing so, students in APE practicums are more likely to be motivated and then fully engaged. As described in the findings, continuous interaction and guidance from APE instructors are necessary to allow undergraduate students to become more reflective and critical when they have practicum settings.

## Figures and Tables

**Table 1 ijerph-20-04282-t001:** Demographic Information of Study Participants.

Participant	Position(s)	Years	Faculty Experience	Concentration
1	Full Professor (Previous department chair)	11	HBCU	APE PETE
2	Associate professor	16	HBCU	APE PETE
3	Adjunct Professor (Retired) (Previous full professor)	29	HBCU, PWI	Motor Development PETE
4	Full Professor (previous department chair)	25	HBCU, PWI	APE PETE
5	Associate Professor (Interim department chair)	10	HBCU, PWI	Athletic Training Exercise Science

PETE = Physical Education Teacher Education; PWI = Predominately White Institution; HBCU = Historically Black Colleges and Universities.

**Table 2 ijerph-20-04282-t002:** The points to be considered and supporting comments for practical experience of APE.

Subthemes	General Description	Rationale	Indicative Quote and/or Supporting Comments
Quality of quantity	The quality of practical experience beyond fulfilling requirements	Meaningful experience of student during practical experience	“It is more about how than what”
	The importance of supervision (Basic instructor roles: regular check-ups, constructive feedback, and clear expectations)	To promote learning of students and develop professionalism.	“Less hours with more supervision will be better for quality of hands-on experience” “One time go and one time reflection time”
	Practicum placement in diverse settings	Developmental engagement into various k-12 settings (e.g., elementary and secondary level) instead of attending in a single place. Importance of the systematic participation in practical experience during four years at college (e.g., APE, method courses, and internships)	“The only hours required in PE is the state requirement for student teaching. There is no requirement before those hours. SHAPE wants, when they actually did recognition, they want you to have an experience at the middle school, or high school, and one in elementary.”
	Consideration of institutional contexts (e.g., transportation, and regional characteristics such as urban and suburban)	To overcome environmental barriers	“Here this is probably one modification I have made I have done ten because transportation is a problem…If we had transportation, or close that they could walk to, then I would probably up it to 15 or twenty hours”
Need for diversity in practical experiences	Available options for practical experience	Instructor’s role to find out the best options in limited circumstances (e.g., regional networks)	“So when you are observing you are still limited because that class is blended with, it may be a regular Physical Education teacher that makes modification for one or two students within a larger class. But it really depends on the school district”
	Complicated situations of practical experience (e.g., influence of class and school district)	Importance of careful choices in advance before students engage in practicum settings	
	Various places (e.g., school size, different locations, diverse groups in terms of race/ethnicity)	Students can broaden their perspective (toward working with various groups of people with special needs)	“Large schools, small schools, and diverse school settings in terms of race and ethnicity. All the different experiences. Students broaden their perspective more and more”
	Consideration of major and curriculum of students	Give a choice matching the major (e.g., community-based settings for exercise science major students, k-12 public schools for PETE major students)	“I give them a choice… So because they are not all going to be teachers”
Practical experience pertaining to APE course	Practical experience as a supplemental part to strengthen student learning as a part of APE course	Importance of APE course itself APE is introductory and a new process for students in general	“And then just really filling that course with a lot of examples, discussions and hands-on experiences about experience where you are being prepared in your program to help them really see how this is really just very different from regular physical education or regular course that they may teach because there is so many different factors that have to be considered with Adapted Physical Education that’s new”
	Providing quality of practical and/or hands-on experience embedded in APE course	Within given options, students can have practical experience of APE in different ways Students can have meaningful practical learning opportunities throughout the course itself	“Hands-on experiences are needed, sometimes it is really hard though to find places where they can observe or work or volunteer and be a part of the school or program where they actually get hands-on experiences. But that’s why we have all those projects in our classes…”
	Solution to compensate existing curricular limitations (e.g., unrelatedness of prerequisite course(s) for APE course)	Introductory course required for various curriculums.	“But the pre-requisites that prepare students for this class is not actually related to modification as far as disabilities in most of them. So this may be the first time where they know that they body changes but of course there are so many disabilities” “The connection between institutional context and APE courses because the curriculum of each program will have a different emphasis like therapeutic recreation, education, or rehabilitation”.
	Continuous communication and frequent feedback	To strengthen student knowledge and critical thinking	“I think they also need to have time to reflect, and to learn how to reflect. So I do a lot of that on discussion boards where they need to interact with each other”

## Data Availability

Not applicable.

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
