# Peer review of "Faculty Perspective Regarding Practical Experience of Adapted Physical Education for Undergraduate Students"

_ijerph, 2023, doi:10.3390/ijerph20054282_

Round 1

Reviewer 1 Report

Thank you for allowing me to provide comments on this manuscript.

Throughout the manuscript, there are misspelled words and grammatical errors. The entire manuscript needs to be carefully checked. Here are a few examples (this list is not complete):

Line 25, spell out the meaning of PETE.

Line 31: "ins APE course" needs to be corrected. I do not know what the authors intended here.

Line 49: "teachers to not" needs to be corrected.

Line 104: "There is no little attention..." needs to be corrected.

Line 121: "had working experience" needs to be corrected.

Line 225: "more supervise will" needs to be corrected.

Line 324: "school" is not spelled correctly (two times)

Line 426: "pseudonym" is not spelled correctly

etc. Again, this list is not complete. Please carefully read through to catch all errors.

The presentation of the results should be improved. It was difficult to read through all of these comments - it is overwhelming to the reader. Consider to creatively put the themes and supporting comments into a table - perhaps one theme per table? Could some of your take-aways be presented with bullets in a table? Think about how best to present this to the reader and at the same time keeping their interest. 

I believe there should be limitations of the study mentioned as well. I didn't see this. The small number of faculty participants is one limitation. And, where do the readers see future studies being helpful surrounding this topic?

The discussion at times feels like it is a repeat of the results. Please review and tweak the discussion with this in mind. 

Thank you. 

Author Response

Editor

- My responses are in blue

- All changes in the revised manuscript are in red

- I use “->” to show the changes between before the revision and after the revision in the manuscript.

Reviewer 1

Comment 1

Thank you for allowing me to provide comments on this manuscript. Throughout the manuscript, there are misspelled words and grammatical errors. The entire manuscript needs to be carefully checked. Here are a few examples (this list is not complete):

Response

Thank you for the constructive feedback. I sincerely appreciate it. I provided a point-by-point response with corrections and changes integrated into the updated manuscript in response to your comments. I hope my responses are satisfactory to you.

I reviewed the whole manuscript and then corrected all errors. I thought that there were no errors because I got a proofreading prior to submission to the journal. However, your feedback really helps me to improve the quality of this article. Sincerely appreciate it again.

Comment 2

Line 26, spell out the meaning of PETE.

Response

Please see line 8, ‘physical education teacher education (PETE)’

Comment 3

Line 37: "ins APE course" needs to be corrected. I do not know what the authors intended here.

Response

Thank you very much for your feedback. I corrected typos and errors like the below.   

>“ins APE course”

Comment 4

Line 50: "teachers to not" needs to be corrected.

Response

>Teachers do not have to be a CAPE to teach an APE class.

Comment 5

Line 105: "There is no little attention..." needs to be corrected.

Response

>“There is a lack of attention”

Comment 6

Line 125: "had working experience" needs to be corrected.

Response

>I corrected this part as “have working experience”  

Comment 7

Line 231: "more supervise will" needs to be corrected.

Response

> I corrected this part as “more supervision will”

Comment 8

"school" is not spelled correctly (two times)

Response

>I corrected school (typo error) as “school”.

Comment 9

Line 435: "pseudonym" is not spelled correctly.

Response

>Thank you for your feedback. I corrected this misspell as “pseudonym”.

Comment 10

etc. Again, this list is not complete. Please carefully read through to catch all errors.

Response

Thank you very much for your time and constructive feedback for me. I reviewed the whole manuscript and then corrected all errors. I thought that there were no errors because I got a proofreading prior to submission to the journal. However, your feedback really helps me to improve the quality of this article. Sincerely appreciate it again.

Comment 11

The presentation of the results should be improved. It was difficult to read through all of these comments - it is overwhelming to the reader. Consider to creatively put the themes and supporting comments into a table - perhaps one theme per table? Could some of your take-aways be presented with bullets in a table? Think about how best to present this to the reader and at the same time keeping their interest. 

Response

Line 486

I agree on your feedback. I put the table at the end of the results section. Please see the below.

Table 2 included the results of this study to represent the points to be considered when planning and implementing practical experiences of APE. Each subtheme with supporting comments and/or the points was described.    

Table 2. The points to be considered and supporting comments for practical experience of APE

Subthemes

General description

Rationale

Indicative quote and/or supporting comments

Quality of quantity 

The quality of practical experience beyond fulfilling requirements

Meaningful experience of student during practical experience  

“It is more about how than what”

The importance of supervision (Basic instructor roles; regular check-ups, constructive feedback, and clear expectations)

To promote learning of students and develop professionalism. 

“Less hours with more supervision will be better for quality of hands-on experience”

“One time go and one time reflection time”

Practicum placement in diverse settings

Developmental engagement into various k-12 settings (e.g., elementary, and secondary level) instead of attending in a single place. 

Importance of the systematic participation in practical experience during four-years at college (e.g., APE, method courses and internships)

“The only hours required in PE is the state requirement for student teaching. There is no requirement before those hours. SHAPE wants, when they actually did recognition, they want you to have an experience at the middle school, or high school, and one in elementary.”

Consideration of institutional contexts (e.g., transportation, and regional characteristics such as urban and suburban)

To ovecome environmental barriers

“Here this is probably one modification I have made I have done ten because transportation is a problem…If we had transportation, or close that they could walk to, then I would probably up it to 15 or twenty hours”

Need for diversity in practical experiences 

Available options for practical experience

Instructor’s role to find out the best options in limited circumstances (e.g., regional networks)

“So when you are observing you are still limited because that class is blended with, it may be a regular Physical Education teacher that makes modification for one or two students within a larger class. But it really depends on the school district”

Complicated situations of practical experience

(e.g., influence of class, and school district)

Importance of careful choices in advance before students engage in practicum settings

Various places (e.g., school size, different locations, diverse groups in terms of race/ethnicity)

>students can broaden their perspective ((toward working with (various group of people with special needs)))

“Large schools, small schools, and diverse school settings in terms of race and ethnicity. All the different experiences. Students broaden their perspective more and more”

Consideration of major and curriculum of students

Give a choice matching to the major (e.g., community-based settings for exercise science major students, k-12 public schools for PETE major students)

“I give them a choice… So because they are not all going to be teachers”

Practical experience pertaining to APE course

Practical experience as a supplemental part to strengthen student learning as a part of APE course   

Importance of APE course itself

APE is introductory and a new process for students in general

“And then just really filling that course with a lot of examples, discussions and hands-on experiences about experience where you are being prepared in your program to help them really see how this is really just very different from regular physical education or regular course that they may teach because there is so many different factors that have to be considered with Adapted Physical Education that’s new”

Providing quality of practical and/or hands-on experience embedded in APE course

Within given options, students can have practical experience of APE in different ways

Students can have meaningful practical learning opportunities throughout the course itself  

“Hands-on experiences are needed, sometimes it is really hard though to find places where they can observe or work or volunteer and be a part of the school or program where they actually get hands-on experiences. But that’s why we have all those projects in our classes…” 

Solution to compensate existing curricular limitations (e.g., unrelatedness of prerequisite course(s) for APE course)

Introductory course required for various curriculums.

“But the pre-requisites that prepare students for this class is not actually related to modification as far as disabilities in most of them. So this may be the first time where they know that they body changes but of course there are so many disabilities”

The connection between institutional context and APE courses because the curriculum of each program will have a different emphasis like therapeutic recreation, education, or rehabilitation.

Continuous communication and frequent feedback 

To strenghen student knolwedge and critical thinking

“I think they also need to have time to reflect, and to learn how to reflect. So I do a lot of that on discussion boards where they need to interact with each other”

Comment

I believe there should be limitations of the study mentioned as well. I didn't see this. The small number of faculty participants is one limitation. And, where do the readers see future studies being helpful surrounding this topic? The discussion at times feels like it is a repeat of the results. Please review and tweak the discussion with this in mind

Response

Thank you for feedback. I included limitations in the conclusion section.

>Line 551

4. Limitations and Future Directions

This study has several limitations. There should be a caution to interpret the results of this study, as there were small size of the sample. However, the focus of this qualitative research explored to gain an in-depth understanding of practical experience of APE course from view of faculty in the context of higher education institutions in the U.S. Other limitation is that two study participants had their expertise in the areas of PETE (motor development), and exercise science (athletic training), respectively. The researcher included these two participants by considering their diverse faculty experience. For instance, these two participants included several topics about individuals with disabilities in their curriculums. Furthermore, they provided practical experiences and/or hands-on experience for undergraduate students towards working with individuals with disabilities. Lastly, although focus of this study was based on the view of faculty, to investigate perspective of diverse stakeholders of APE (e.g., parents or guardians of individuals with disabilities, in-service and/or pre-service teachers) will be needed in that there is a limited number of these lines of inquiries. Also, future study should consider more contextual factors such as the types of disabilities, and diverse age groups connecting to APE practicum settings.

Reviewer 2 Report

INTRODUCTION

The author conducted qualitative research on the topic of adapted physical education (APE). The topic is well known in the literature, with a number of recent studies. Therefore, the author aims to publish a relevant study that fits into the scientific discourse. The findings provide a relevant contribution to the field. 

STRENGTHS

The structure of the study is logical, orderly and follows the structure expected for empirical studies. The single figure presented in the study is appropriate and interpretable, the study does not cite itself.

The strengths of the study are that the author uses empirically sound methodology, proceeds with care, and bases his findings on expert opinion. The conclusions are accurate and relevant. The introductory chapter presents the topic in an interesting, detailed and well-supported way. The sampling procedure is appropriate to the research objectives and the research meets ethical standards.

The study is recommended for publication with minor modifications.

WEAKNESSES

The following recommendations were made during the review:

1. References cited are not always recent and they are not comprehensive

Most of the bibliography contains studies published in international scientific journals. Relevant literature is cited where appropriate, but in many cases the more recent literature contains less recent literature, also listed as a separate source.

For example, literature number 1 refers to literature numbers 2 and 3. The reference to references 2 and 3 in the text is relevant, but it is worth considering whether redundant literature should be included at all.

In addition, references 33-35 are included in section "2.4 Data analysis and Trustworthiness". In this case, the bibliography of reference 34 includes reference 33 (dated 1989) and the inclusion of the older work was considered redundant during the review. Furthermore, reference 35 does not appear to be relevant to the claim after which it appears. In conclusion, in this case it may have been sufficient to include reference 33 supporting the statement in question.

An example of incomprehensiveness is that the study includes two studies from 2007, even though new studies by the same well-known expert are published frequently, most recently in 2021.

Authors should formally check the bibliography and, if necessary, adapt it to the requirements of the journal (e.g. book titles in italics, etc.): https://www.mdpi.com/authors/references  

Rather than providing further examples, the author is encouraged to check the references in the light of the above.

2. The wording of the research questions does not seem coherent to the reviewer, which is also related to the methodology used

At the end of the introductory chapter, the author formulates two interview questions. It was not clear to the reviewer how and why these questions were related to the interview topics described in chapter '2.3 Data collection'. If the latter three are procedural sub-questions/sub-themes, it is suggested that the two questions in the introductory chapter should be called research questions rather than interview questions. This is related to the fact that the same questions are referred to by the author in chapter 'Discussion 4' as 'primary questions guiding this research' rather than as questions asked of respondents.

After reflection and examination, the connection between these is understandable, but I suggest that the author make the connection easier for the reader to understand and that they use more precise nomenclature.

It is also recommended that the first research question be reformulated. Questions of the type "which variables influence what" attract a non-qualitative methodology. Qualitative research is better suited to questions that begin with "how", as the author has done in the second research question.

2.1 Methodological issues related to the research questions

The author defined the research and interview questions deductively. However, in chapter 2.4 'Data analysis and Trustworthiness' it is described that aspects were extracted from the interview texts, which is an inductive method ('Common themes and sub-themes emerged from the data'). If the author analysed the data inductively, i.e. created categories and subcategories (grounded theory?) and then tested the resulting categories with the research subjects, it is suggested that the relationship between the two logical paths be clarified.

The author does not tell the reader how the data were coded. Did he use software or did he code freely by hand?

Although the author did not calculate a reliability index for the subcategories, he describes that he checked the categories and results with the research participants, which is consistent with qualitative methodology.

3. Discussion and conclusion

If the author used an inductive method for his/her research, a presentation of the new theory is expected. The discussion section is of sufficient length and relevance, but also contains many findings that should be included in the conclusion section. In this context, sentences 2 and 4 of the conclusion section refer to literature that provides information about the topic of the research, not the results of the research.

I suggest moving the findings that contribute to the theory from the discussion chapter to the conclusion chapter and I suggest formulating the new theory there. Also, I do not recommend including additional information or references in the conclusion chapter, except in very justified cases.

Author Response

Editor

- My responses are in blue

- All changes in the revised manuscript are in red

- I use “->” to show the changes between before the revision and after the revision in the manuscript.

Reviewer 2

Comment 1

INTRODUCTION

The author conducted qualitative research on the topic of adapted physical education (APE). The topic is well known in the literature, with a number of recent studies. Therefore, the author aims to publish a relevant study that fits into the scientific discourse. The findings provide a relevant contribution to the field. 

STRENGTHS

The structure of the study is logical, orderly and follows the structure expected for empirical studies. The single figure presented in the study is appropriate and interpretable, the study does not cite itself.

The strengths of the study are that the author uses empirically sound methodology, proceeds with care, and bases his findings on expert opinion. The conclusions are accurate and relevant. The introductory chapter presents the topic in an interesting, detailed and well-supported way. The sampling procedure is appropriate to the research objectives and the research meets ethical standards.

The study is recommended for publication with minor modifications.

Response

Thank you for your kind comments, and the constructive feedback. I sincerely appreciate it. I provided a point-by-point response with corrections and changes integrated into the updated manuscript in response to your comments. I hope my responses are satisfactory to you.

Comment 2

The following recommendations were made during the review:

1. References cited are not always recent and they are not comprehensive

Most of the bibliography contains studies published in international scientific journals. Relevant literature is cited where appropriate, but in many cases the more recent literature contains less recent literature, also listed as a separate source. For example, literature number 1 refers to literature numbers 2 and 3. The reference to references 2 and 3 in the text is relevant, but it is worth considering whether redundant literature should be included at all.

Response

Thank you so much for your feedback. I carefully reviewed and then corrected the references in the manuscript.

>When I conducted a literature review, I tried to search the most recent literature published within 10 years as possible. However, if there is not many and/or no information, I tried to see the studies published after 2000.

>As you indicated, there were many and inappropriate or redundant literature cited. Based on your feedback, I reviewed the whole manuscript and then edited reference information. I reflected by myself and went back to the most important basic step to organize better for my manuscript. Sincerely appreciate your feedback (Originally there was 42 references. After I reviewed the manuscript, 8 references deleted). Also, I tried to put the most recent and related references at most).

Comment 3

In addition, references 33-35 are included in section “2.4 Data analysis and Trustworthiness". In this case, the bibliography of reference 34 includes reference 33 (dated 1989) and the inclusion of the older work was considered redundant during the review. Furthermore, reference 35 does not appear to be relevant to the claim after which it appears. In conclusion, in this case it may have been sufficient to include reference 33 supporting the statement in question. An example of incomprehensiveness is that the study includes two studies from 2007, even though new studies by the same well-known expert are published frequently, most recently in 2021.

Authors should formally check the bibliography and, if necessary, adapt it to the requirements of the journal (e.g. book titles in italics, etc.): https://www.mdpi.com/authors/references  

Rather than providing further examples, the author is encouraged to check the references in the light of the above.

Response

Thank you very much for your feedback.

>As you indicated, in section 2.4., it is appropriate to include only reference instead of including unnecessary references (33, & 35). I corrected the references accordingly. 

> Connecting to the previous Comment 3, as you indicated, there were redundant literature cited. Based on your feedback, I reviewed the whole manuscript and then updated reference information for this study. Sincerely appreciate your valuable feedback.

Line 203

To ensure trustworthiness of the findings, the current study established the following four criteria: credibility, transferability, dependability, and confirmability - which are basic components of qualitative research [29] [33-35].

Comment 4

2. The wording of the research questions does not seem coherent to the reviewer, which is also related to the methodology used

At the end of the introductory chapter, the author formulates two interview questions. It was not clear to the reviewer how and why these questions were related to the interview topics described in chapter '2.3 Data collection'. If the latter three are procedural sub-questions/sub-themes, it is suggested that the two questions in the introductory chapter should be called research questions rather than interview questions. This is related to the fact that the same questions are referred to by the author in chapter 'Discussion 4' as 'primary questions guiding this research' rather than as questions asked of respondents.

After reflection and examination, the connection between these is understandable, but I suggest that the author make the connection easier for the reader to understand and that they use more precise nomenclature. It is also recommended that the first research question be reformulated. Questions of the type "which variables influence what" attract a non-qualitative methodology. Qualitative research is better suited to questions that begin with "how", as the author has done in the second research question.

Response

Thank you for your feedback. As you indicated, there were two research questions at introductory part.  Latter three questions should be called as procedural sub-questions. Accordingly, I changed Section 4. Also, I agree on your comment that research questions should be begin with ‘how’ rather than using ‘what’ in qualitative research. I changed first research question at introductory part. Thank you for your feedback. Please see the below.

Line 115

Before

Specifically, interview participants were asked following questions: What are the variables that impact planning and implementing practical experiences in APE courses for undergraduate students?

After

Specifically, there were two research questions: how various factors influence on the planning and implementing practical experiences in APE courses for undergraduate students?

Line 179

Before

>Interviews included three guiding topics;

After

Interview questions were related to three guiding topics;

Line 492

Before

> The primary questions guiding this research were: What are the variables that impact planning and implementing practical experiences in APE courses for undergraduate students?

After

There were two research questions guiding this research: how various factors influence on the planning and implementing practical experiences in APE courses for undergraduate students?

Comment 5   

2.1 Methodological issues related to the research questions.

The author defined the research and interview questions deductively. However, in chapter 2.4 'Data analysis and Trustworthiness' it is described that aspects were extracted from the interview texts, which is an inductive method ('Common themes and sub-themes emerged from the data'). If the author analyzed the data inductively, i.e. created categories and subcategories (grounded theory?) and then tested the resulting categories with the research subjects, it is suggested that the relationship between the two logical paths be clarified.

Response

Thank you so much for your feedback. I agree that this study was based on inductive methods. Data of the study participants through the interviews were analyzed by the researcher’s initial coding to discern the concepts within the data. Those concepts were rearranged to identify common and sub-themes using axial coding. Each code was checked from the interview transcripts.

I tried to collect data from lived experience of study participants. And then I coded and categorized the data so that I could make themes and subthemes. So, I believe this method is the same as grounded theory. I tried to understand better about practical experience for APE based on the experienced faculty view because there were not many of even no study to see faculty view. I started this study hoping that I would provide a kind of new approach or information for the quality of APE practical experience.

As you pointed out, it is important to make logical connection between research questions and methodology. Sincerely appreciate your feedback. Connecting to Comment 4, first research question was changed based on your feedback which is better suited for this type of qualitative research. 

However, as you gave me feedback, I need to include information regarding a new kind of theory or any explanation in the conclusion section. I added several sentences in the conclusion section based on the findings. Please see the below (Comment 8 and response).

Comment 6  

The author does not tell the reader how the data were coded. Did he use software or did he code freely by hand? Although the author did not calculate a reliability index for the subcategories, he describes that he checked the categories and results with the research participants, which is consistent with qualitative methodology.

Response  

>Thank you for your feedback. I did not use software for coding for this study. However, I personally feel that I need to be familiar with the qualitative research tools such as MAXQDA so that I could work more effectively regarding conceptualization of texts, rearrangements, and other works like making a figure and/or chart.

> Personally, I am trying to conduct research works and practices one thing at a time as a rookie faculty (for instance, I learned and used Rstudio for my other quantitative research recently and this work was published yesterday!!). Another qualitative work that accepted in other journal (Taylor and Francis; this will be published soon), it was my first qualitative research experience as the main author, I worked with my committee (expertise in qualitative methods (a senior professor) and his advisee (= current assistant professor at Maryland; this advisee was my colleague at the Ohio State University in the U.S). I could learn the complexity of qualitative research by collaborating with them but could not have an opportunity to master about the software).

>Based on your feedback, I will make sure to use the software for my future study. I believe this will increase my research productivity and also make my research works in more robust ways in terms of long term. Sincerely appreciate it. Instead, I tried to create table 2 to represent the results of this study dividing into subthemes.

Comment 7

3. Discussion and conclusion

If the author used an inductive method for his/her research, a presentation of the new theory is expected. The discussion section is of sufficient length and relevance, but also contains many findings that should be included in the conclusion section. In this context, sentences 2 and 4 of the conclusion section refer to literature that provides information about the topic of the research, not the results of the research.

Response

Thank you for your feedback. I reviewed the discussion and conclusion section. As you indicated, sentence 2 and 4 of the conclusion section were already explained in the introductory section. So I deleted these two sentences. Also, connecting to Comment 8, I edited the conclusions section by removing additional information/references.

Comment 8

I suggest moving the findings that contribute to the theory from the discussion chapter to the conclusion chapter and I suggest formulating the new theory there. Also, I do not recommend including additional information or references in the conclusion chapter, except in very justified cases.   

Response 8-1

Thank you for the constructive feedback. Your feedback really helped me to improve overall quality of the manuscript. Sincerely appreciate it! As you indicated, I moved the findings in the discussion section into the conclusion section. Also, I reviewed and edited the conclusion section as well based on your feedback. Please see the below.  

Before

4. Discussion

The current study explored the perspective of faculty with relation to practical experiences in an APE course for undergraduate students in kinesiology and PETE programs. The primary questions guiding this research were: What are the variables that impact planning and implementing practical experiences in APE courses for undergraduate students? How do faculty effectively create practical experiences in an APE course to enhance undergraduate students’ knowledge and understanding of working with populations with disabilities?There were themes that emerged from the participant interviews; (a) Quality of Quantity, (b) Need for Diversity in Practical Experiences and (c) Practical Experience pertaining to APE course. The findings align with the previous literature which states practical experiences are an integral part for student learning in terms of their pedagogical practice and professional preparation [2,5,6,7,8,11,25,36,37]. The findings of this study further explain how undergraduate students in APE courses could maximize their learning through practical experiences of APE course. It is not enough to merely fulfil practicum hours as a requirement. Faculty believed the quality of practical experiences may be more important than the overall number of hours spent in practical experiences. There is very little criteria for student practicum hours or details specifically for APE courses, however, faculty indicated more diverse settings would certainly be beneficial. Particularly, based on the national standards (e.g., SHAPE America) it is recommended that undergraduate students should be exposure to diverse settings in terms or prek-12 and diverse leaners before student teaching because it will lead students being more competitive and prepared [36-39]. The faculty emphasis on the need for diverse practical experiences in this study was consistent with previous literature on practical experiences in APE courses [2,4].

Furthermore, for practical experiences to be effective and of high quality, faculty have an important role in improving students’ learning during the process of practical experiences. From the very beginning, instructors must provide clear expectations for students about what is expected in terms of their professional behavior and engagement in practical settings [40]. In the same context, it is necessary to have active communication with students so that faculty can provide corrective feedback and then check their student’s learning which could result in more successful and meaningful practical experiences. These findings were consistent with previous studies which shed light on the role of active interaction with students during their practicum and course completion at college level [38,41]. 

Contextual factors are another important variable that impact practical experiences. There are variables which could impact the options for practical experiences in the APE course. For instance, the structure and physical environment could be a decisive factor. It could be appropriate to give a choice to students for practical experiences in the context of their specialization and program because there can be several different areas of focus in programs within the area of kinesiology.

Also, more diverse practical experiences would be beneficial for student learning. The options available to students for practical experiences will vary depending on location, physical environment, and other contextual factors of the institution. For instance, variables such as lack of transportation and limited adapted physical sports and activities available in the community would negatively impact the over practical experience. Faculty also indicated the importance of networking in the community to find the best options for practical experiences. Faculty indicated making a bridge for practical experience through APE lectures will be one effective way.    

With relation to the second question how faculty effectively create practical experiences in an APE course to enhance undergraduate students’ knowledge and understanding of working with populations with disabilities, there were several findings. Depending on the ways in which practical experiences were organized as a part of the APE course, faculty in this study believed their students would experience professional growth. As discussed in the above, faculty had a crucial role for practical experiences. For this reason, it could be beneficial for students if they have on campus practical experiences for APE courses. Especially, if students in the APE course have barriers to attending practical experiences, such as transportation issues and lack of options on the community settings. There would be additional benefits because students in APE courses could learn each topic then connect it to weekly practicum in sequential ways. Students in the APE practicum can be paired with individuals with a disability. They could learn how to plan and implement an activity or lesson based on the specific type of disability. However, diverse options for practical experiences both on and off campus would still be beneficial for student learning [1,4]. Campus based APE practicum is a self-contained environment. However, undergraduate students also need to experience inclusive settings because they can strengthen their knowledge pertaining to working with individuals with disabilities and become aware of real word settings by going to community-based schools and facilities [2,4,37].

Generally, undergraduate students only take one APE course to learn how to work with populations with special needs in PE and physical activity settings. Students are likely to lack understanding and be hesitated to work with populations with disabilities [16,26]. Therefore, APE courses should include a depth of information regarding the disabilities students may encounter. Once a foundation of knowledge is provided at an introductory level, students can be gradually introduced to more intermediate levels of learning through diverse hands-on experiences. This approach will allow students to have repeated mastery as well as enable instructors to observe the progress of students. Lastly, there can be several difference foci within kinesiology and PETE programs. Therefore, instructors of APE courses should consider the specific program focus to determine which experiences will be most meaningful and related to their specific major. By doing so, students in APE practicum are more likely to be motivated and then fully engaged [11,36]. As described in the findings, continuous interaction and guidance from APE instructors is necessary to allow undergraduate students to become more reflective and critical when they have practicum settings [25,36,42].

After

4. Discussion

The current study explored the perspective of faculty with relation to practical experiences in an APE course for undergraduate students in kinesiology and PETE programs. There were two research questions guiding this research: How various factors influence on the planning and implementing practical experiences in APE courses for undergraduate students? and How do faculty effectively create practical experiences in an APE course to enhance undergraduate students’ knowledge and understanding of working with populations with disabilities?There were themes that emerged from the participant interviews; (a) Quality of Quantity, (b) Need for Diversity in Practical Experiences and (c) Practical Experience pertaining to APE course. The findings align with the previous literature which states practical experiences are an integral part for student learning in terms of their pedagogical practice and professional preparation [1,7,20-24]. Particularly, based on the national standards (e.g., SHAPE America) it is recommended that undergraduate students should be exposure to diverse settings in terms or prek-12 and diverse leaners before student teaching because it will lead students being more competitive and prepared [18].  The faculty emphasis on the need for diverse practical experiences in this study was consistent with previous literature on practical experiences in APE courses [23,34].

Furthermore, for practical experiences to be effective and of high quality, faculty have an important role in improving students’ learning during the process of practical experiences. From the very beginning, instructors must provide clear expectations for students about what is expected in terms of their professional behavior and engagement in practical settings [34]. These findings were consistent with previous studies which shed light on the role of active interaction with students during their practicum and course completion at college level [30-31]. Contextual factors are another important variable that impact practical experiences. There are variables which could impact the options for practical experiences in the APE course. For instance, the structure and physical environment could be a decisive factor. Also, more diverse practical experiences would be beneficial for student learning. The options available to students for practical experiences will vary depending on location, physical environment, and other contextual factors of the institution. For instance, variables such as lack of transportation and limited adapted physical sports and activities available in the community would negatively impact the over practical experience.

With relation to the second question how faculty effectively create practical experiences in an APE course to enhance undergraduate students’ knowledge and understanding of working with populations with disabilities, there were several findings. Depending on the ways in which practical experiences were organized as a part of the APE course, faculty in this study believed their students would experience professional growth. As discussed in the above, faculty had a crucial role for practical experiences. It could be beneficial for students if they have on campus practical experiences for APE courses if students in the APE course have barriers to attending practical experiences, such as transportation issues and lack of options on the community settings. There would be benefits because students in APE courses could learn each topic and then connect it to weekly practicum in sequential ways. Students in the APE practicum can be paired with individuals with a disability. For instance, they could learn how to plan and implement an activity or lesson based on the specific type of disability. However, diverse options for practical experiences both on and off campus would still be beneficial for student learning [1,4]. Campus based APE practicum is a self-contained environment. However, undergraduate students also need to experience inclusive settings because they can strengthen their knowledge pertaining to working with individuals with disabilities and become aware of real word settings by going to community-based schools and facilities [1,12].

Generally, undergraduate students only take one APE course to learn how to work with populations with special needs in PE and physical activity settings. Students are likely to lack understanding and be hesitated to work with populations with disabilities [32-35]. Therefore, APE courses should include a depth of information regarding the disabilities students may encounter. Once a foundation of knowledge is provided at an introductory level, students can be gradually introduced to more intermediate levels of learning through diverse hands-on experiences. This approach will allow students to have repeated mastery as well as enable instructors to observe the progress of students.

Before

Conclusions

The current study explored the ways in which practical experiences in APE courses could be effectively conducted for practical learning of undergraduate students in the areas of kinesiology and PETE programs. Study participants described that quality of practical experiences could be improved through faculty’s active communication with students to provide feedback, support and encourage reflection [43]. There is no doubt that faculty believed pre-service teachers and practitioners can enhance their learning through diverse experiences. APE courses are generally more introductory in nature and only one course is required during the curriculum of undergraduate students in kinesiology and PETE programs, making practicum experiences critical for student learning [14-16].

It could be appropriate to give a choice to students for practical experiences in the context of their specialization and program because there can be several different areas of focus in programs within the area of kinesiology. By doing so, students in APE practicum are more likely to be motivated and then fully engaged [11,36]. As described in the findings, continuous interaction and guidance from APE instructors is necessary to allow undergraduate students to become more reflective and critical when they have practicum settings [25,36,42].

After

Conclusions

The current study explored the ways in which practical experiences in APE courses could be effectively conducted for practical learning of undergraduate students in the areas of kinesiology and PETE programs. There is no doubt that faculty believed pre-service teachers and practitioners can enhance their learning through diverse experiences. The findings of this study further explain how undergraduate students in APE courses could maximize their learning through practical experiences of APE course. It is not enough to merely fulfil practicum hours as a requirement. The quality of practical experiences may be more important than the overall number of hours spent in practical experiences. There is very little criteria for student practicum hours or details specifically for APE courses, however, faculty indicated more diverse settings would certainly be beneficial. The findings explained the role of instructors in more specific ways. It is necessary to have active communication with students so that faculty can provide corrective feedback and then check their student’s learning which could result in more successful and meaningful practical experiences. Faculty also indicated the importance of networking in the community to find the best options for practical experiences.

Contextual factors such as physical environment could be a decisive factor which could impact the options for practical experiences in the APE course. It could be appropriate to give a choice to students for practical experiences in the context of their specialization and program because there can be several different areas of focus in programs within the area of kinesiology. Faculty indicated making a bridge for practical experience through APE lectures will be one effective way. Lastly, there can be several difference foci within kinesiology and PETE programs. Therefore, instructors of APE courses should consider the specific program focus to determine which experiences will be most meaningful and related to their specific major. By doing so, students in APE practicum are more likely to be motivated and then fully engaged. As described in the findings, continuous interaction and guidance from APE instructors is necessary to allow undergraduate students to become more reflective and critical when they have practicum settings.

Response 8-2

I agree that I need add new explanation (and/or a new theory) in that this study used grounded theory as a qualitative method. In the conclusion section, I added new explanations based on the findings of this study (sincerely appreciate your feedback!). By following your guidance and comments, I could add a kind of new explanation in the conclusions section (Even though I did not propose a new theory (I really thought for one week about a new theory that I could add. I could add new explanations).

I hope these new explanations could be appropriate for the manuscript. By conducting future works, I could try to make a kind of new explanation and then propose a new theory sometime. By conducting this study, I could set up a line of this inquiry in several ways. Sincerely appreciate it.

Reviewer 3 Report

First, I appreciate reading this interesting article. The author's suggested ways for students to repeat their studies and instructors observing students' progress were quite impressive. 

The overall paper's structure was well-written. However, I want to point out some points about this research.

First, only demographic information is described in the table. Since data analysis was applied to this approach, I hope some tables or figures also depict the results, especially data analysis results. 

Furthermore, I hope the paper has some related research about this topic. 

Author Response

 Editor

- My responses are in blue

- All changes in the revised manuscript are in red

- I use “->” to show the changes between before the revision and after the revision in the manuscript.

Thank you for the constructive feedback. I sincerely appreciate it. I provided a point-by-point response with corrections and changes integrated into the updated manuscript in response to your comments. I hope my responses are satisfactory to you.

Reviewer 3

Comment 1

First, I appreciate reading this interesting article. The author's suggested ways for students to repeat their studies and instructors observing students' progress were quite impressive. 

Response

Thank you for the constructive feedback. I sincerely appreciate it. I provided a point-by-point response with corrections and changes integrated into the updated manuscript in response to your comments. I hope my responses are satisfactory to you.

Comment 2  

The overall paper's structure was well-written. However, I want to point out some points about this research. First, only demographic information is described in the table. Since data analysis was applied to this approach, I hope some tables or figures also depict the results, especially data analysis results. Furthermore, I hope the paper has some related research about this topic. 

Response

Line 486

I agree on your feedback. I put the table at the end of the results section. Please see the below.

Table 2 included the results of this study to represent the points to be considered when planning and implementing practical experiences of APE. Each subtheme with supporting comments and/or the points was described.    

Table 2. The points to be considered and supporting comments for practical experience of APE

Subthemes

General description

Rationale

Indicative quote and/or supporting comments

Quality of quantity 

The quality of practical experience beyond fulfilling requirements

Meaningful experience of student during practical experience  

“It is more about how than what”

The importance of supervision (Basic instructor roles; regular check-ups, constructive feedback, and clear expectations)

To promote learning of students and develop professionalism. 

“Less hours with more supervision will be better for quality of hands-on experience”

“One time go and one time reflection time”

Practicum placement in diverse settings

Developmental engagement into various k-12 settings (e.g., elementary, and secondary level) instead of attending in a single place. 

Importance of the systematic participation in practical experience during four-years at college (e.g., APE, method courses and internships)

“The only hours required in PE is the state requirement for student teaching. There is no requirement before those hours. SHAPE wants, when they actually did recognition, they want you to have an experience at the middle school, or high school, and one in elementary.”

Consideration of institutional contexts (e.g., transportation, and regional characteristics such as urban and suburban)

To ovecome environmental barriers

“Here this is probably one modification I have made I have done ten because transportation is a problem…If we had transportation, or close that they could walk to, then I would probably up it to 15 or twenty hours”

Need for diversity in practical experiences 

Available options for practical experience

Instructor’s role to find out the best options in limited circumstances (e.g., regional networks)

“So when you are observing you are still limited because that class is blended with, it may be a regular Physical Education teacher that makes modification for one or two students within a larger class. But it really depends on the school district”

Complicated situations of practical experience

(e.g., influence of class, and school district)

Importance of careful choices in advance before students engage in practicum settings

Various places (e.g., school size, different locations, diverse groups in terms of race/ethnicity)

>students can broaden their perspective ((toward working with (various group of people with special needs)))

“Large schools, small schools, and diverse school settings in terms of race and ethnicity. All the different experiences. Students broaden their perspective more and more”

Consideration of major and curriculum of students

Give a choice matching to the major (e.g., community-based settings for exercise science major students, k-12 public schools for PETE major students)

“I give them a choice… So because they are not all going to be teachers”

Practical experience pertaining to APE course

Practical experience as a supplemental part to strengthen student learning as a part of APE course   

Importance of APE course itself

APE is introductory and a new process for students in general

“And then just really filling that course with a lot of examples, discussions and hands-on experiences about experience where you are being prepared in your program to help them really see how this is really just very different from regular physical education or regular course that they may teach because there is so many different factors that have to be considered with Adapted Physical Education that’s new”

Providing quality of practical and/or hands-on experience embedded in APE course

Within given options, students can have practical experience of APE in different ways

Students can have meaningful practical learning opportunities throughout the course itself  

“Hands-on experiences are needed, sometimes it is really hard though to find places where they can observe or work or volunteer and be a part of the school or program where they actually get hands-on experiences. But that’s why we have all those projects in our classes…” 

Solution to compensate existing curricular limitations (e.g., unrelatedness of prerequisite course(s) for APE course)

Introductory course required for various curriculums.

“But the pre-requisites that prepare students for this class is not actually related to modification as far as disabilities in most of them. So this may be the first time where they know that they body changes but of course there are so many disabilities”

The connection between institutional context and APE courses because the curriculum of each program will have a different emphasis like therapeutic recreation, education, or rehabilitation.

Continuous communication and frequent feedback 

To strenghen student knolwedge and critical thinking

“I think they also need to have time to reflect, and to learn how to reflect. So I do a lot of that on discussion boards where they need to interact with each other”

Reviewer 4 Report

Introduction and theoretical framework

There is inconsistency between the ways of citing, either use numbers according to the order of appearance, or use the APA norms (see line 32), but not both.

Between lines 51 and 64 there are missing references.

Design

It is recommended not to use pseudonyms, and to use numbers for the different subjects.

The methodology is well designed in general

Results

A summary table (more quantitative) of the typology of responses and their categorization should be included, in order to make inferences or extrapolate the results to other studies, or an image for the interpretation of the results.

Discussion and conclusions

They are correct and appropriate references are used

bibliographic references

Are relevant to the subject of the study

Author Response

 Editor

- My responses are in blue

- All changes in the revised manuscript are in red

- I use “->” to show the changes between before the revision and after the revision in the manuscript.

Thank you for the constructive feedback. I sincerely appreciate it. I provided a point-by-point response with corrections and changes integrated into the updated manuscript in response to your comments. I hope my responses are satisfactory to you.

Reviewer 4

Comment 1

Introduction and theoretical framework

There is inconsistency between the ways of citing, either use numbers according to the order of appearance, or use the APA norms (see line 32), but not both.

Between lines 51 and 64 there are missing references.

Response

Thank you for your feedback. I aware that there are two parts to use the APA norms and the order of appearance for citation(s) at the same time. I agree on your feedback that it would be better to have consistency. Thus, I changed several parts based on your feedback. Please see the below.

I reviewed the whole manuscript and then corrected all errors about references. I thought that there were no errors because I got a proofreading prior to submission to the journal. However, your feedback really helps me to improve the quality of this article. Sincerely appreciate it again!

>Line 33

Before

Perlman & Piletic (2012) emphasized the importance of improving pedagogical knowledge of undergraduate students throughout practical experiences of APE course at universities [11].

After

Practical experience of APE course is an integral part to enhance pedagogical knowledge of undergraduate students [11].  

>Line 99

Before

In a recent review by Case et al (2021), the authors described those practical experiences in the community setting, with numerous self-choice options were most beneficial for changing attitudes of undergraduate students toward population with disabilities

After

In a recent meta-analysis review, the authors described those practical experiences in the community setting, with numerous self-choice options were most beneficial for changing attitudes of undergraduate students toward population with disabilities    

Comment 2

Design - It is recommended not to use pseudonyms, and to use numbers for the different subjects. The methodology is well designed in general.

Response

Based on your feedback, I corrected several parts. Also I updated table 1. Please see the below.

>Line 141

Before

Table 1. Demographic Information of Study Participants.

Pseudonym

Position(s)

Years

Faculty Experience

Concentration

Kios

Full Professor

11

HBCU

APE

PETE

Avira

Department Chair

(previous years)

16

HBCU

APE

PETE

Alex

Vice Chair

29

HBCU

Motor Development

PETE

Nunes

Full Professor

Department Chair

25

HBCU

APE

PETE

Marie

Associate Professor

Interim Department Chair

10

HBCU

Athletic Training Exercise Science

After

Table 1. Demographic Information of Study Participants.

Participant

Position(s)

Years

Faculty Experience

Concentration

1

Full Professor

(Previous department chair)

11

HBCU

APE

PETE

2

Associate professor

16

HBCU

APE

PETE

3

Adjunct Professor (Retired) 

(Previous full professor)

29

HBCU, PWI  

Motor Development

PETE

4  

Full Professor

(previous department chair)

25

HBCU, PWI

APE

PETE

5  

Associate Professor

(Interim department chair)

10

HBCU, PWI

Athletic Training Exercise Science

PETE = Physical Education Teacher Education; PWI = Predominately White Institution; HBCU = Historically Black Colleges and Universities

>Line 136

Before

Online video meetings were preferred for two participants (Drs. Avira & Alex). Other participants had in-person interviews by maintaining social distancing and complying with COVID-19 health policies of their respective institutions (Drs. Kios, Nunes and Marie).

After

Online video meetings were preferred for two participants (Participant 2, and 3). Other participants had in-person interviews by maintaining social distancing and complying with COVID-19 health policies of their respective institutions (Participant 1, 4, and 5).

>Line 153-171: I changed pseudonyms as numbers like the below.

Before

… One participant (Dr. Alex) is currently working as adjunct professor since she retired. Each participant had different work experiences. Three faculty members (Drs. Alex, Nunes and Marie) had full time faculty working experience at both PWI and HBCUs, and two participants have worked in HBCUs (Drs. Avira and Kios) …

After

… One participant (3) is currently working as adjunct professor since she retired. Each participant had different work experiences. Three faculty members (3, 4, & 5) had full time faculty working experience at both PWI and HBCUs, and two participants have worked in HBCUs (1, & 2) …

>Line 216-484 (Results section)

I changed all the pseudonyms as numbers in the manuscript.

Comment 3

Results - A summary table (more quantitative) of the typology of responses and their categorization should be included, in order to make inferences or extrapolate the results to other studies, or an image for the interpretation of the results.

Response

Line 486

I agree on your feedback. I put the table at the end of the results section. Please see the below.

Table 2 included the results of this study to represent the points to be considered when planning and implementing practical experiences of APE. Each subtheme with supporting comments and/or the points was described.    

Table 2. The points to be considered and supporting comments for practical experience of APE

Subthemes

General description

Rationale

Indicative quote and/or supporting comments

Quality of quantity 

The quality of practical experience beyond fulfilling requirements

Meaningful experience of student during practical experience  

“It is more about how than what”

The importance of supervision (Basic instructor roles; regular check-ups, constructive feedback, and clear expectations)

To promote learning of students and develop professionalism. 

“Less hours with more supervision will be better for quality of hands-on experience”

“One time go and one time reflection time”

Practicum placement in diverse settings

Developmental engagement into various k-12 settings (e.g., elementary, and secondary level) instead of attending in a single place. 

Importance of the systematic participation in practical experience during four-years at college (e.g., APE, method courses and internships)

“The only hours required in PE is the state requirement for student teaching. There is no requirement before those hours. SHAPE wants, when they actually did recognition, they want you to have an experience at the middle school, or high school, and one in elementary.”

Consideration of institutional contexts (e.g., transportation, and regional characteristics such as urban and suburban)

To ovecome environmental barriers

“Here this is probably one modification I have made I have done ten because transportation is a problem…If we had transportation, or close that they could walk to, then I would probably up it to 15 or twenty hours”

Need for diversity in practical experiences 

Available options for practical experience

Instructor’s role to find out the best options in limited circumstances (e.g., regional networks)

“So when you are observing you are still limited because that class is blended with, it may be a regular Physical Education teacher that makes modification for one or two students within a larger class. But it really depends on the school district”

Complicated situations of practical experience

(e.g., influence of class, and school district)

Importance of careful choices in advance before students engage in practicum settings

Various places (e.g., school size, different locations, diverse groups in terms of race/ethnicity)

>students can broaden their perspective ((toward working with (various group of people with special needs)))

“Large schools, small schools, and diverse school settings in terms of race and ethnicity. All the different experiences. Students broaden their perspective more and more”

Consideration of major and curriculum of students

Give a choice matching to the major (e.g., community-based settings for exercise science major students, k-12 public schools for PETE major students)

“I give them a choice… So because they are not all going to be teachers”

Practical experience pertaining to APE course

Practical experience as a supplemental part to strengthen student learning as a part of APE course   

Importance of APE course itself

APE is introductory and a new process for students in general

“And then just really filling that course with a lot of examples, discussions and hands-on experiences about experience where you are being prepared in your program to help them really see how this is really just very different from regular physical education or regular course that they may teach because there is so many different factors that have to be considered with Adapted Physical Education that’s new”

Providing quality of practical and/or hands-on experience embedded in APE course

Within given options, students can have practical experience of APE in different ways

Students can have meaningful practical learning opportunities throughout the course itself  

“Hands-on experiences are needed, sometimes it is really hard though to find places where they can observe or work or volunteer and be a part of the school or program where they actually get hands-on experiences. But that’s why we have all those projects in our classes…” 

Solution to compensate existing curricular limitations (e.g., unrelatedness of prerequisite course(s) for APE course)

Introductory course required for various curriculums.

“But the pre-requisites that prepare students for this class is not actually related to modification as far as disabilities in most of them. So this may be the first time where they know that they body changes but of course there are so many disabilities”

The connection between institutional context and APE courses because the curriculum of each program will have a different emphasis like therapeutic recreation, education, or rehabilitation.

Continuous communication and frequent feedback 

To strenghen student knolwedge and critical thinking

“I think they also need to have time to reflect, and to learn how to reflect. So I do a lot of that on discussion boards where they need to interact with each other”

Reviewer 5 Report

The study examines a current issue and can be considered filling a gap, as it examines APE practices from the institution's perspective. The study could provide a good foundation for a later, larger-scale research, which could also present the perspectives of other participants (students, practicing teachers, children and families).

It would be advisable to include also in the abstract how many interviews were conducted.

The exploration of the literature background is thorough and to the point, furthermore, fits well with the examined topic.The sentences that can be read after the presentation of the theoretical background (lines 109-115) would fit better in chapter 2.

A research based on five interviews is more suitable for raising questions and for outlining the problem, moreover, the participants do not adequately represent the field's educators (e.g. they are mostly from HBCUs). For this reason, the results and conclusions are of limited validity, this should be included in the “Limitations”.

The presentation of the participants could be given a clearer format (even by expanding the previous table).

The methodological description is detailed and thorough.

The presentation of the results is organized around three topics, during which the author supports his analysis with quotes from the interview texts. Sometimes the amount of citations is more than the author's analysis. Since there are only five interviews, the results could have been supplemented by the use of content analysis software.

The discussion reflects well on the results and contains many useful ideas that can serve as a starting point for further research.

I think that this study should be considered more of a problem-revealing nature, since it is not possible to provide adequate answers to the research questions based on results of five interviews. These methodological limitations must be mentioned as limitation.

Typing errors:

line 149: full professor

line 324: black school

Author Response

 Editor

- My responses are in blue

- All changes in the revised manuscript are in red

- I use “->” to show the changes between before the revision and after the revision in the manuscript.

Thank you for the constructive feedback. I sincerely appreciate it. I provided a point-by-point response with corrections and changes integrated into the updated manuscript in response to your comments. I hope my responses are satisfactory to you.

Reviewer 5

Comment 1

The study examines a current issue and can be considered filling a gap, as it examines APE practices from the institution's perspective. The study could provide a good foundation for a later, larger-scale research, which could also present the perspectives of other participants (students, practicing teachers, children and families).

Response

Thank you for the constructive feedback. I sincerely appreciate it. I provided a point-by-point response with corrections and changes integrated into the updated manuscript in response to your comments. I hope my responses are satisfactory to you.

Comment 2

It would be advisable to include also in the abstract how many interviews were conducted.

Response

Thank you for your feedback. I included number of participants information in the abstract.

Line 12

“There were five study participants in this study”

Comment 3  

The exploration of the literature background is thorough and to the point, furthermore, fits well with the examined topic. The sentences that can be read after the presentation of the theoretical background (lines 109-115) would fit better in chapter 2.

Response

Thank you very much. I agree on your feedback. I moved this part into the first part of chapter 2 (materials and methods, line 113-119).

Comment 4  

A research based on five interviews is more suitable for raising questions and for outlining the problem, moreover, the participants do not adequately represent the field's educators (e.g. they are mostly from HBCUs). For this reason, the results and conclusions are of limited validity, this should be included in the “Limitations”.

Response

Thank you for feedback. I included limitations in the conclusion section.

>Line 551-567

4. Limitations and Future Directions

This study has several limitations. There should be a caution to interpret the results of this study, as there were small size of the sample. However, the focus of this qualitative research explored to gain an in-depth understanding of practical experience of APE course from view of faculty in the context of higher education institutions in the U.S. Other limitation is that two study participants had their expertise in the areas of PETE (motor development), and exercise science (athletic training), respectively. The researcher included these two participants by considering their diverse faculty experience. For instance, these two participants included several topics about individuals with disabilities in their curriculums. Furthermore, they provided practical experiences and/or hands-on experience for undergraduate students towards working with individuals with disabilities. Lastly, although focus of this study was based on the view of faculty, to investigate perspective of diverse stakeholders of APE (e.g., parents or guardians of individuals with disabilities, in-service and/or pre-service teachers) will be needed in that there is a limited number of these lines of inquiries. Also, future study should consider more contextual factors such as the types of disabilities, and diverse age groups connecting to APE practicum settings.

Comment 5

The presentation of the participants could be given a clearer format (even by expanding the previous table).

Response

Sincerely appreciate your feedback. Based on you and other reviewer’s comment, I reviewed and then updated table 1 to provide better format. Please see line 145-174 (I tried to provide thick description of study participants to consider this study is based on qualitative study).

Also, I changed study participant’s information by numbering them instead of using pseudonyms. This would be a clearer format for readers. Please see the below.

>Line 141

Before

Table 1. Demographic Information of Study Participants.

Pseudonym

Position(s)

Years

Faculty Experience

Concentration

Kios

Full Professor

11

HBCU

APE

PETE

Avira

Department Chair

(previous years)

16

HBCU

APE

PETE

Alex

Vice Chair

29

HBCU

Motor Development

PETE

Nunes

Full Professor

Department Chair

25

HBCU

APE

PETE

Marie

Associate Professor

Interim Department Chair

10

HBCU

Athletic Training Exercise Science

After

Table 1. Demographic Information of Study Participants.

Participant

Position(s)

Years

Faculty Experience

Concentration

1

Full Professor

(Previous department chair)

11

HBCU

APE

PETE

2

Associate professor

16

HBCU

APE

PETE

3

Adjunct Professor (Retired) 

(Previous full professor)

29

HBCU, PWI  

Motor Development

PETE

4  

Full Professor

(previous department chair)

25

HBCU, PWI

APE

PETE

5  

Associate Professor

(Interim department chair)

10

HBCU, PWI

Athletic Training Exercise Science

PETE = Physical Education Teacher Education; PWI = Predominately White Institution; HBCU = Historically Black Colleges and Universities

>Line 136

Before

Online video meetings were preferred for two participants (Drs. Avira & Alex). Other participants had in-person interviews by maintaining social distancing and complying with COVID-19 health policies of their respective institutions (Drs. Kios, Nunes and Marie).

After

Online video meetings were preferred for two participants (Participant 2, and 3). Other participants had in-person interviews by maintaining social distancing and complying with COVID-19 health policies of their respective institutions (Participant 1, 4, and 5).

>Line 153-171: I changed pseudonyms as numbers like the below.

Before

… One participant (Dr. Alex) is currently working as adjunct professor since she retired. Each participant had different work experiences. Three faculty members (Drs. Alex, Nunes and Marie) had full time faculty working experience at both PWI and HBCUs, and two participants have worked in HBCUs (Drs. Avira and Kios) …

After

… One participant (3) is currently working as adjunct professor since she retired. Each participant had different work experiences. Three faculty members (3, 4, & 5) had full time faculty working experience at both PWI and HBCUs, and two participants have worked in HBCUs (1, & 2) …

>Line 216-484 (Results section)

I changed all the pseudonyms as numbers in the manuscript.

Comment 6   

The methodological description is detailed and thorough. The presentation of the results is organized around three topics, during which the author supports his analysis with quotes from the interview texts. Sometimes the amount of citations is more than the author's analysis. Since there are only five interviews, the results could have been supplemented by the use of content analysis software.

Response

Line 486

I agree on your feedback. I put the table at the end of the results section. Please see the below.

Table 2 included the results of this study to represent the points to be considered when planning and implementing practical experiences of APE. Each subtheme with supporting comments and/or the points was described.    

Table 2. The points to be considered and supporting comments for practical experience of APE

Subthemes

General description

Rationale

Indicative quote and/or supporting comments

Quality of quantity 

The quality of practical experience beyond fulfilling requirements

Meaningful experience of student during practical experience  

“It is more about how than what”

The importance of supervision (Basic instructor roles; regular check-ups, constructive feedback, and clear expectations)

To promote learning of students and develop professionalism. 

“Less hours with more supervision will be better for quality of hands-on experience”

“One time go and one time reflection time”

Practicum placement in diverse settings

Developmental engagement into various k-12 settings (e.g., elementary, and secondary level) instead of attending in a single place. 

Importance of the systematic participation in practical experience during four-years at college (e.g., APE, method courses and internships)

“The only hours required in PE is the state requirement for student teaching. There is no requirement before those hours. SHAPE wants, when they actually did recognition, they want you to have an experience at the middle school, or high school, and one in elementary.”

Consideration of institutional contexts (e.g., transportation, and regional characteristics such as urban and suburban)

To ovecome environmental barriers

“Here this is probably one modification I have made I have done ten because transportation is a problem…If we had transportation, or close that they could walk to, then I would probably up it to 15 or twenty hours”

Need for diversity in practical experiences 

Available options for practical experience

Instructor’s role to find out the best options in limited circumstances (e.g., regional networks)

“So when you are observing you are still limited because that class is blended with, it may be a regular Physical Education teacher that makes modification for one or two students within a larger class. But it really depends on the school district”

Complicated situations of practical experience

(e.g., influence of class, and school district)

Importance of careful choices in advance before students engage in practicum settings

Various places (e.g., school size, different locations, diverse groups in terms of race/ethnicity)

>students can broaden their perspective ((toward working with (various group of people with special needs)))

“Large schools, small schools, and diverse school settings in terms of race and ethnicity. All the different experiences. Students broaden their perspective more and more”

Consideration of major and curriculum of students

Give a choice matching to the major (e.g., community-based settings for exercise science major students, k-12 public schools for PETE major students)

“I give them a choice… So because they are not all going to be teachers”

Practical experience pertaining to APE course

Practical experience as a supplemental part to strengthen student learning as a part of APE course   

Importance of APE course itself

APE is introductory and a new process for students in general

“And then just really filling that course with a lot of examples, discussions and hands-on experiences about experience where you are being prepared in your program to help them really see how this is really just very different from regular physical education or regular course that they may teach because there is so many different factors that have to be considered with Adapted Physical Education that’s new”

Providing quality of practical and/or hands-on experience embedded in APE course

Within given options, students can have practical experience of APE in different ways

Students can have meaningful practical learning opportunities throughout the course itself  

“Hands-on experiences are needed, sometimes it is really hard though to find places where they can observe or work or volunteer and be a part of the school or program where they actually get hands-on experiences. But that’s why we have all those projects in our classes…” 

Solution to compensate existing curricular limitations (e.g., unrelatedness of prerequisite course(s) for APE course)

Introductory course required for various curriculums.

“But the pre-requisites that prepare students for this class is not actually related to modification as far as disabilities in most of them. So this may be the first time where they know that they body changes but of course there are so many disabilities”

The connection between institutional context and APE courses because the curriculum of each program will have a different emphasis like therapeutic recreation, education, or rehabilitation.

Continuous communication and frequent feedback 

To strenghen student knolwedge and critical thinking

“I think they also need to have time to reflect, and to learn how to reflect. So I do a lot of that on discussion boards where they need to interact with each other”

Comment 7

The discussion reflects well on the results and contains many useful ideas that can serve as a starting point for further research. I think that this study should be considered more of a problem-revealing nature, since it is not possible to provide adequate answers to the research questions based on results of five interviews. These methodological limitations must be mentioned as limitation.

Response

>Line 551

4. Limitations and Future Directions

This study has several limitations. There should be a caution to interpret the results of this study, as there were small size of the sample. However, the focus of this qualitative research explored to gain an in-depth understanding of practical experience of APE course from view of faculty in the context of higher education institutions in the U.S. Other limitation is that two study participants had their expertise in the areas of PETE (motor development), and exercise science (athletic training), respectively. The researcher included these two participants by considering their diverse faculty experience. For instance, these two participants included several topics about individuals with disabilities in their curriculums. Furthermore, they provided practical experiences and/or hands-on experience for undergraduate students towards working with individuals with disabilities. Lastly, although focus of this study was based on the view of faculty, to investigate perspective of diverse stakeholders of APE (e.g., parents or guardians of individuals with disabilities, in-service and/or pre-service teachers) will be needed in that there is a limited number of these lines of inquiries. Also, future study should consider more contextual factors such as the types of disabilities, and diverse age groups connecting to APE practicum settings.

Comment 7   

Typing errors:

line 149: full professor

line 338: black school

Response

Thank you very much for your time and constructive feedback for me. I reviewed the whole manuscript and then corrected all errors. I thought that there were no errors because I got a proofreading prior to submission to the journal. However, your feedback really helps me to improve the quality of this article. Sincerely appreciate it again.

>Line 149: full professor

I checked and the corrected any typing errors.

>Line 338: black school

Based on your feedback, I tried to change this part in more appropriate ways. To clarify the intention of interviewee. I corrected “black school” as “diverse school settings in terms of race and ethnicity”.

Before

Large school, small schools, white school, black school

After

Large schools, small schools, and diverse school settings in terms of race and ethnicity
